# Large local variations in the use of health services in rural southern Ethiopia: An ecological study

Hiwot Abera Areru [1,2] *, Mesay Hailu Dangisso[1], Bernt Lindtjørn[1,2]

**1** School of Public Health, College of Medicine and Health Sciences, Hawassa University, Hawassa, Ethiopia, **2** Global Public Health and Primary Care, Centre for International Health, University of Bergen, Bergen, Norway

☯ These authors contributed equally to this work.

* hiwotab_2005@yahoo.com

**Data Availability Statement:** We have deposited the dataset used for this manuscript in a repository called Zenodo. The URL and DOI of the dataset can be found from "Hiwot Abera Areru. (2021). Health

## Abstract

Ethiopia is behind schedule in assuring accessible, equitable and quality health services. Understanding the geographical variability of the health services and adjusting small-area level factors can help the decision-makers to prioritize interventions and allocate scarce resources. There is lack of information on the degree of variation of health service utilisation at micro-geographic area scale using robust statistical tools in Ethiopia. Therefore, the objective of this study was to assess the health service utilisation and identify factors that account for the variation in health service utilisation at kebele (the smallest administrative unit) level in the Dale and Wonsho districts of the Sidama region. An exploratory ecological study design was employed on the secondary patient data collected from 1 July 2017 to 30 June 2018 from 65 primary health care units of the fifty-four kebeles in Dale and Wonsho districts, in the Sidama region. ArcGIS software was used to visualise the distribution of health service utilisation. SaTScan analysis was performed to explore the unadjusted and covariate-adjusted spatial distribution of health service utilisation. Linear regression was applied to adjust the explanatory variables and control for confounding. A total of 67,678 patients in 54 kebeles were considered for spatial analysis. The distribution of the health service utilisation varied across the kebeles with a mean of 0.17 visits per person per year (Range: 0.01–1.19). Five kebeles with health centres had a higher utilisation rate than other rural kebeles without health centres. More than half (57.4%) of the kebeles were within a 10 km distance from health centres. The study found that distance to the health centre was associated with the low health care utilisation. Improving the accessibility of health services by upgrading the primary health care units could increase the service use.

## Introduction

Globally, in the last three decades, considerable improvements have been observed in most public health interventions, but with substantial regional variability [1]. Ethiopia is making an

service use_spatial_ dataset [Data set]. Zenodo. https://doi.org/10.5281/zenodo.5213926".

**Funding:** Norwegian Programme for Capacity Development in Higher Education and Research for Development/ South Ethiopia Network of Universities in Public Health (NORHED/SENUPH) project provided funding in the form of a grant awarded to BL (ETH-13/0025). URL: https://www.uib.no/en. The sponsors or funders have no role in the study design, data collection and analysis, decision to publish, or preparation of the manuscript.

**Competing interests:** The authors have declared that no competing interests exist.

increasing effort to improve the health services [2], yet remains behind schedule on meeting Sustainable Development Goals (SDG), like SDG 3, which is about healthy lives and well-being [3]. Sustainable Development Goal 3 aims to improve the health and well-being of the population through universal health coverage [4]. Universal health coverage encompasses the provision of accessible, equitable and quality health services without incurring financial hardship [5].

In Ethiopia, primary health care coverage is used as a proxy indicator for service access. Primary health coverage is defined as access to primary health care (health centres and health posts) in relation to the total population [6]. Service access consists availability, affordability, and acceptability, accommodation, affordability and acceptability dimensions [7, 8]. Health care utilization is the description or quantification of health care service usage [9]. R. Andersen proposed a behavioural health service use model with major components that influence the health outcome. Population factors were considered as major predictors for health service use and health outcomes. These factors include predisposing (demographic, health beliefs, values, attitude, and knowledge), enabling (availability of health service, income, travelling and waiting time) and need factors. The other determinants in the model were the health care system and external environmental factors. Health service use was considered as an outcome on previous version of the model, but it is considered as a determinant of subsequent health outcome in the latest version [10, 11].

Major indicators used for quantification of health service utilisation in Ethiopia include outpatient attendance per individual, rate of admission, rate of bed occupancy and average length of stay [12]. According to the national criteria, areas within 2 hours of walking distance or less than 10 km away from the primary health care unit are considered physically accessible [6]. A study conducted 30 years ago in 26 health facilities in central, southern and western Ethiopia, showed that substantial variations in health service utilisation rates occurred in regions, district, and local communities. Furthermore, the geographic determinants for the health service utilisation rate were physical accessibility, such as the presence of all-weather roads, size and location of the health facilities, and distance of households away from the health facilities, distribution of other health services in the area, population density and urbanisation [13]. However, the Federal Ministry of Health launched roadmap in 2020 which was designed to improve the coverage, access, and quality of primary health care units. The health posts and health centres will be upgraded. According to the Ethiopian three-tire health service delivery system, one health centre serves 25,000 rural population up to 40,000 urban population. There are five community health posts under each health centre. Each health posts serve 3,000–50000 population. Five health posts and one referral health centre comprise a primary health care unit [14, 15]. The scale-up was initiated to tackle the challenges observed in the health extension program in addressing universal health coverage and delivering quality services. The upgraded health facilities will be equipped with better health professionals, clinical services, infrastructure, governance and leadership, and information systems [15–17]. In this new plan, health posts that are far (>1 hour) from health centres, and in geographically inaccessible areas, will be upgraded to "comprehensive health posts". In addition to the basic health extension packages, the comprehensive health posts will provide improved clinical and curative services. During this 15 years plan (2020–2035), a health centre and health post will be merged if they are close to each other. These improvements could increase the coverage, accessibility, quality and service usage [17].

Geographical analyses are used for variables having a spatial structure and it is helpful in public health decision-making by illustrating health events and health services [18]. Thus, Geographic Information System (GIS) could be used by health planners to examine the level of health service utilisation and to visualize over-and underutilised areas. Different studies

showed a variation in health care utilisation patterns from place to place even at the small-area level [19]. Even if primary health services are available, they are usually underutilized [20].

Studies elsewhere in Ethiopia, showed that accessibility of the health facilities was better for urban residents and for those with shortest distance to the nearest health facilities [21, 22]. However, the geographical distributions of health services, disparities in health service utilisation at the kebele (smallest administrative units) level have not been investigated. Moreover, understanding geographical factors contributing to uneven distribution of health service utilisation at a small area (kebele) level might help to plan targeted intervention. We also integrated the determining factors in the context of the behavioural model of health service use by Ronald M. Andersen (S1 Fig). Specific questions we seek to investigate are: (1) How is health service utilisation distributed at kebele level? (2) What kinds of kebele level geographic factors affect the service usage? The answer to these questions might give direction to policy-makers on improving the health service accessibility. Therefore, this study aims to assess the health service utilisation and identify geographical factors contributing to the low service usage in the Dale and Wonsho districts of the Sidama region in southern Ethiopia.

## Materials and methods

### Ethics statement

Ethical clearance was obtained from the Institutional Review Board at the College of Medicine and Health Sciences of Hawassa University (Reference number IRB/022/10) and the Regional Ethics Committee for Medical and Health Research in Norway REK Vest (2018/67/REK vest). The then Sidama Zone Health Department, Dale and Wonsho woreda (district) health offices provided permission letters to collect secondary data from all health facilities. The ethical committees mentioned above ruled that no informed consent was required for secondary data use. All the methods used in this study were according to relevant guidelines and regulations. There was no personal identifier in the data set. The confidentiality of the data was maintained.

### Study area, setting and design

This study was conducted in the Sidama region in southern Ethiopia. The region has three agro-ecologic or climatic zones, such as wet or moist highland areas (with altitudes above 3,001 metres), semi-arid midland areas (with altitudes ranging from 2,001 metres to 3,000 metres) and dry lowland areas (1,501 metres to 2,000) [23, 24]. The Sidama region is one of the most densely populated areas in Ethiopia, with a total area of 6981 km$^2$ and a population density of 533 persons/km$^2$ [23]. The region constitutes about 4.0% of the national population, 95% of the population lives in rural areas and 94% of the population speak the local language called "*Sidaamu Afoo*". Females compose 49.5% of the population. Under-five years of age children constitute 15.8% of the population [25]. Based on the regional reports, pneumonia was considered the most common cause of morbidity in under-five children [23]. An earlier study from southern-central Ethiopia showed malaria as a major perceived cause of death among adults [26].

The livelihood of the population depends on the farming of perennial crops including maize and Enset (*Ensete ventricosum*) as a staple food and coffee as a cash crop. These crops constitute more than half of the backyard gardening areas and the share of other produces, like maize, khat (*Catha edulis*), fruits, vegetables, pulses, roots, tubers and spices is smaller. Even though Enset plays a pivotal role in Sidama culture and identity, recently the society is changing to maize farming [23, 27, 28]. The home garden of the Sidama people also includes trees and livestock species, such as cattle, goats, sheep, donkeys, horses, mules and chickens. The

presence of livestock in the home garden shows the integrated farming system in Sidama [27]. The Sidama society is known for its indigenous traditional institutions of conflict resolution mainly due to land issues, but they also make use of the formal government structures, such as courts [29].

The modern health care service started in the region during the late 1930s in Yirga Alem and the use of traditional medicine reduced gradually afterwards [30]. The region had 658 functioning public health facilities (three hospitals, 132 health centres and 523 health posts) and seven non-governmental organization led clinics in 2015 [23].

We did the study in Dale and Wonsho districts (woredas). In 2017, the Dale district had a total population of 268,839 people and an estimated 53,768 households. It had 36 rural and two urban kebeles (the lowest administrative structures). Yirga Alem town was the main town in the district and had five urban kebeles. Wonsho, which was the smallest district in the region had 129,730 people living in 17 rural and one urban kebeles in 21,857 households [23, 31]. There were ten functional public health centres and 33 health posts in Dale, and the Wonsho district had five health centres and 17 health posts. There were nine privately owned primary and medium clinics in Dale district and Yirga Alem town, while only one primary clinic gave service for Wonsho district [23, 31].

This study employed an exploratory ecological study design [32] by using an aggregated data at a population level. This method compares rates in different places to search for spatial patterns that might imply environmental or other etiologic hypotheses. We collected the data from 1 July 2017 to 30 June 2018.

## Study participants

We used all new cases or patients who sought health services from all health centres and health posts in the Dale and Wonsho districts. The national health management information guide indicates to use new cases as a numerator for morbidity calculations [12]. We selected the districts purposively since they were the study sites for the previous demographic study done by the same authors [33]. A total of 67,678 (92%) cases from 73,513 new cases had confirmed addresses [34]. The rest of the cases either didn't have an address or were from other districts. Therefore, we performed the spatial analysis on the 67,678 cases from 54 kebeles having an address at the kebele level.

To confirm the sample adequacy, we performed sample size calculation on Open Epi version 3.01 (Dean AG, Sullivan KM, Soe MM, Open Source Epidemiologic Statistics for Public Health, Version, 2013) statistical software. We used a formula for estimating a single population proportion with a prevalence of 48% utilization, 80% power, 95% confidence interval, a precision of 1% and design effect of 1.5. The required sample size was 14,246 patients. Hence, we have included the 67,678 cases with addresses for the analysis (Fig 1).

## Study variables

The outcome variable was the health service utilisation rate calculated as the number of new visits to the health facilities in the kebele divided by the total population of the kebele. The exposure variables were altitude, population density, family size and distance from the health centre to the centre of the kebele.

## Operational definitions

Health service utilisation rate is the number of new visits to health facilities per year relative to the total population of the same geographical area.

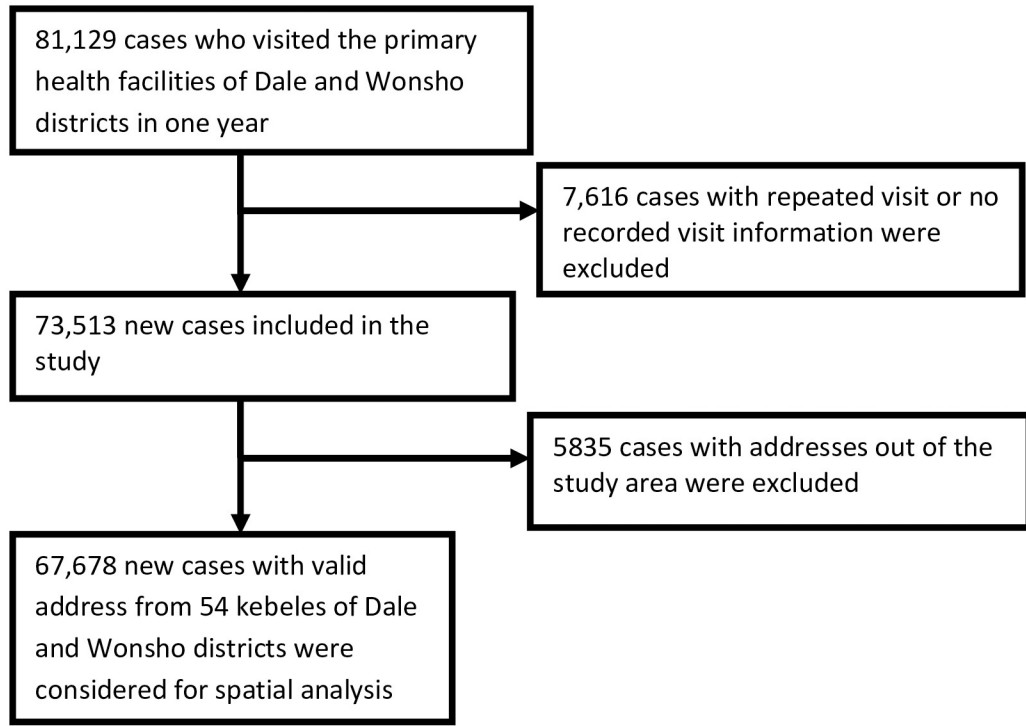

**Fig 1. A flow-chart of number of cases included for the spatial analysis in Dale and Wonsho districts, 2017/18, Sidama, Ethiopia.**

Family size refers to the average number of people living together in a household in a kebele.

Distance from the health centre to the centre of the kebele is the distance measured in kilometres from the health centre in the kebele to the centre of the kebele. It is a proxy indicator for access [12].

### Data collection tools and procedures

The data were collected from registries of 15 health centres and 50 health posts in the Dale and Wonsho districts. Secondary data was collected from the standard registers supplied by the Federal Ministry of Health to the health centres and health posts. Each unit or department had a registry with the registration, identification, and service related information. The information reviewed from the registries is attached in S1 Table. The data collection instruments were pretested on health centres and health posts outside of both districts [34]. Fifteen data collectors were trained for two days about the protocol and how to collect the data from the registries. The data collectors went to the service delivery units with permission letters from each district and copied the information in to the data collection format from the respective registries in the study period. The data were double entered and validated in EpiData version 3.1 software (EpiData Association; Odense, Denmark, 2004). The data cleaning and analysis were done by STATA software version 13 (Stata Corp LP., College Station, Texas, USA, 2013) and Microsoft Excel. Re-checking was done by the principal investigator on 5% of the cases in each health facility to assure the quality of the data. These steps have been described in a previous paper [34].

Geographical information for the study kebeles was collected from different sources. Geographically weighted central locations were indicated by the kebele central locations as coordinates. We extracted the geographical coordinates (latitudes and longitudes) of the kebeles from a previous study done in the region collected by using the geographic positioning system (GPS) tools [35]. These coordinates were used to build the base-maps of the study area and point-maps of the health centres. The Central Statistical Agency of Ethiopia collects geographical data and distributes to the regions. Therefore, population density, family size and altitude information were obtained from the Central Statistical Agency, Hawassa branch, and Sidama Region Plan Commission. The distance information was generated by the ArcGIS 10.3 (Esri Inc., Redlands, CA, USA, 2014) analysis tool by proximity-near command from the kebele central location and health centre location files. We used distance from health centres to the centre of the kebele because our unit of analysis was at kebele level not household level.

## Data analysis

We calculated the health service utilisation rate for each kebele by dividing the number of new visits during one year by the total population in the kebeles using OpenEpi Version 3.01 software.

We made an attribute table having information about the population, kebele codes, number of visits to the health facilities, number of people in each kebele, area coverage of each kebele, health service utilisation rates, population density, family size, altitude, distance from health centres, the coordinates of each kebele and health centres. Then, it was exported to ArcGIS 10.3 for visualization. The World Geodetic System (WGS) 1984, Universal Transverse Mercator (UTM) Zone 37°N was used to define the coordinates' projection. The layer map and the attribute table were joined during the analysis. We used a five scales classification, indicated by high (dark colour) to low (light colour) rates, created with the natural break method for clustering maps. The attribute tables created for ArcGIS was then exported to SaTScan version 9·6·1 software for further analysis of the presence of areas with unusually low rates of health service utilisation or clustering. We used four different types of files for SaTScan analysis. We used the case file, the population file and the coordinate file for initial SaTScan analysis, and covariate files for adjustment analysis. To identify locations and estimate cluster sizes, we used Kulldorf's spatial scan statistics [36]. We assessed the distribution of health centres using the network analyst extension. We prepared a new dataset consisting of the road map, the location of health centres and the central location of the kebeles for network analysis. We assessed service area coverage using a 2 km, 5 km, and 10 km distance away from the facility as a cut of point. Service area coverage is defined in this study as the area measured through physical distance between the service area (health centres) and the people benefiting from the service (kebele centre) [37, 38]. We performed the location-allocation analysis to understand how the kebele centres were within a 10 km distance from the health centres. The Federal Ministry of Health of Ethiopia was used to consider an area to be physically accessible if it is within 10 km distance or 2 hours travel time from the health facility [6]. The setting "maximize coverage" was selected with the fifteen health centre locations.

**Analysis of the geographic grouping of health service utilisation or clustering.** We mapped and labelled the health service utilisation rates in each kebeles using ArcGIS 10.3 (Esri Inc., USA) software. For local clustering analysis, the scan statistics evaluated if the health service utilisation were randomly distributed over a defined area. If the process was not random, the scan statistics helped to identify significant spatial clusters, the log-likelihood ratio (LLR), the relative risk (RR) and P-value. The statistical significance of the largest likelihood ratio was assessed through the Standard Monte Carlo simulation (999 simulations performed). We used a purely spatial Poisson probability model to identify and locate areas with high rates. The

most likely cluster and secondary clusters were identified based on the log-likelihood ratio. No geographic overlap was selected as criteria for reporting secondary clusters. A circular window was used by Kulldorf's spatial scan to identify significant clusters of health service utilisation over the study area. The maximum reported cluster size was set at a radius of less than 50% of the total population at risk. For each scanning window, a likelihood ratio test was conducted to test whether there is an increased rate of health service utilisation as compared with the distribution outside, the window at a P-value <0.05 [19].

**Analysis of determinants of the geographic grouping of health service utilisation.**
After identifying the presence of clustering of health service utilisation rate, we considered the geographical factors as covariates. The geographic grouping observed on health service utilisation might be due to aggregation of known geographic covariates not randomly distributed among the kebeles or due to the presence of spatial dependency. For the Poisson probability model, it is recommended to make the covariates as categorical variables [36], therefore, we categorized the variables accordingly. Then, we adjusted the covariates, such as altitude, population density, and distance from the health centre in the advanced input feature of SaTScan software. We left the variable family size out of the analysis because it was similar in all kebeles. We checked for multicollinearity among exposure variables by using Variance Inflation Factor (VIF). We also performed bivariate and multivariable linear regression to identify the association between the covariates and the outcome (rate of health service utilisation).

We included all three variables in the multivariable regression even if the P-value was >0.2 during bivariate analysis. We used a P-value of 0.05 and the 95% confidence interval as a measure of significance.

Finally, the results of the analyses were presented in tables and on the maps to show the locations where higher rates of service utilisation exist.

## Results

### Spatial distribution of health service utilisation at kebele level

The annual health service utilisation rate per person varied across the kebeles. Bokaso town from Wonsho district showed the highest annual health service utilisation rate 1.19 visits per person per year (95% CI: 1.15 1.23). Masincho kebele from Dale district had the lowest utilisation rate, 0.01 visits per person per year (95% CI: 0.004 0.01). The mean annual health service utilisation rate was 0.17 visits per person per year (S2 Table).

Fig 2 showed four of the five kebeles with the highest utilisation rate had a health centre. These included two kebeles from Wonsho district namely Hunkute and Bokaso urban kebeles; and two kebles from Dale district, Semen Kege and Bua Bedagelo.

### Distribution of health centres

There were 15 health centres in Dale and Wonsho districts during the study period. We demonstrated their distribution and performed a network analysis to identify the areas that were distant from the nearby health centre location based on the road network of the study site. We identified most areas were within a 5 km distance, yet there were areas without health centres located more than 10 km away. The location-allocation analysis showed that 31 out of 54 (57.4%) kebele centres were within a 10 km distance from a health centre (Fig 3 and S2 Fig). This analysis shows the areas that are deprived of nearby health centres in their district.

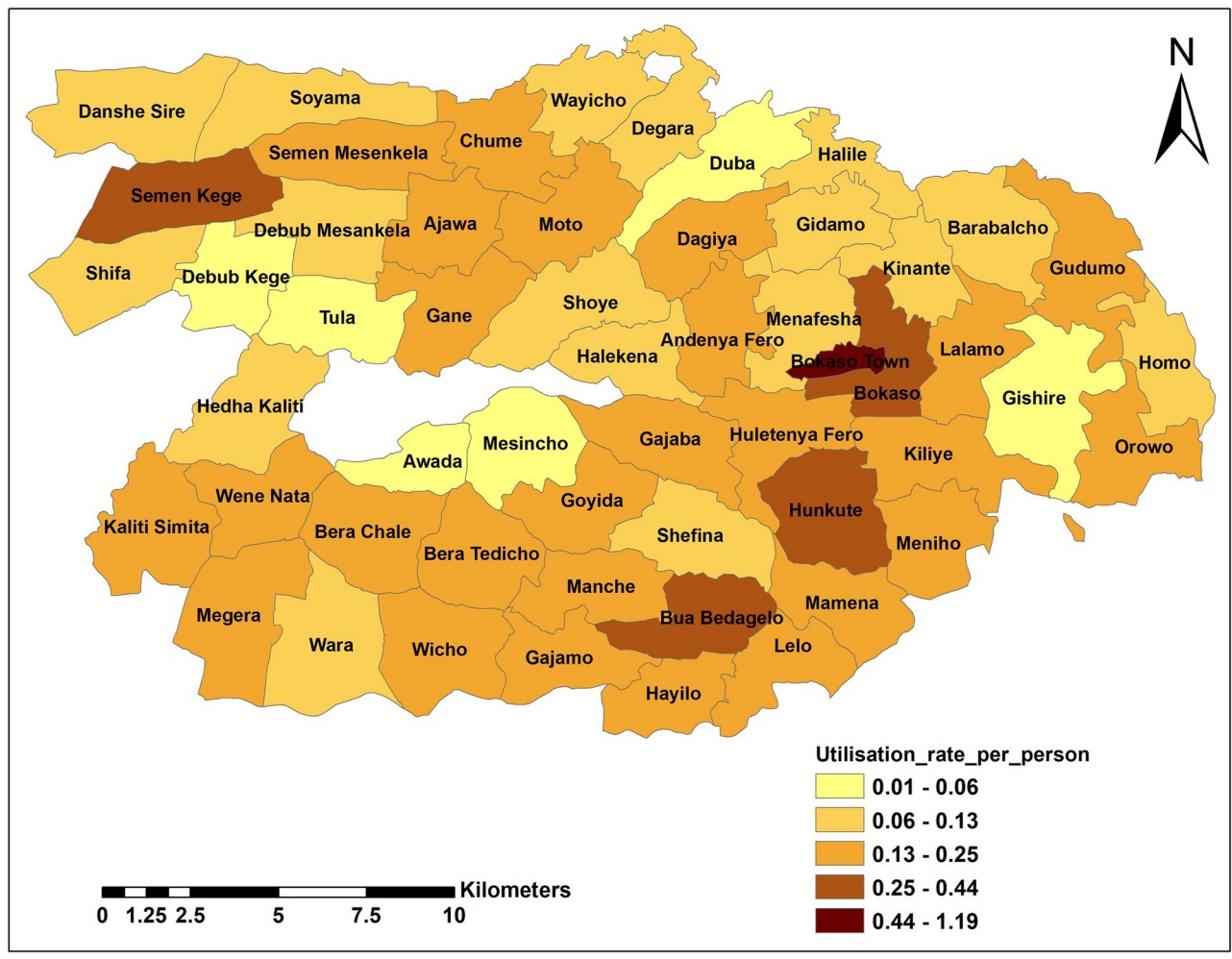

**Fig 2. Spatial distribution of annual health service utilisation rate by kebeles in Dale and Wonsho districts, Sidama, southern Ethiopia, 2017/18 (Unit: Rate per person per year).**

## Purely spatial analysis of health service utilisation

We applied Kulldorff's scan statistics to explore the spatial clustering of health service utilisation. The purely spatial analysis identified significant most-likely cluster and secondary clusters for the low health service utilisation rates. The most-likely cluster revealed the cluster which is least likely to be due to chance. Secondary clusters showed other clusters detected in the data with a p-value <0.05. There were 23 kebeles in the most-likely cluster area for the low health service utilisation (Fig 4 and Table 1).

The relative risk (RR) of the low health service utilisation rate for a most likely cluster area was 0.61 with an observed number of 23572 cases, compared with 31663.5 expected cases. Gishire, Kinante, Barabalcho, Shefina, and Gajamo kebeles were secondary cluster areas. Compared to people living in other kebeles, those living in the most likely cluster window were 0.61 times less likely to use health services (Table 1).

## Relationship of geographic variables with health service utilisation

After including other kebele-level covariates, such as altitude, distance from a health centre and population density, we performed the analysis in SaTScan software. The kebeles included

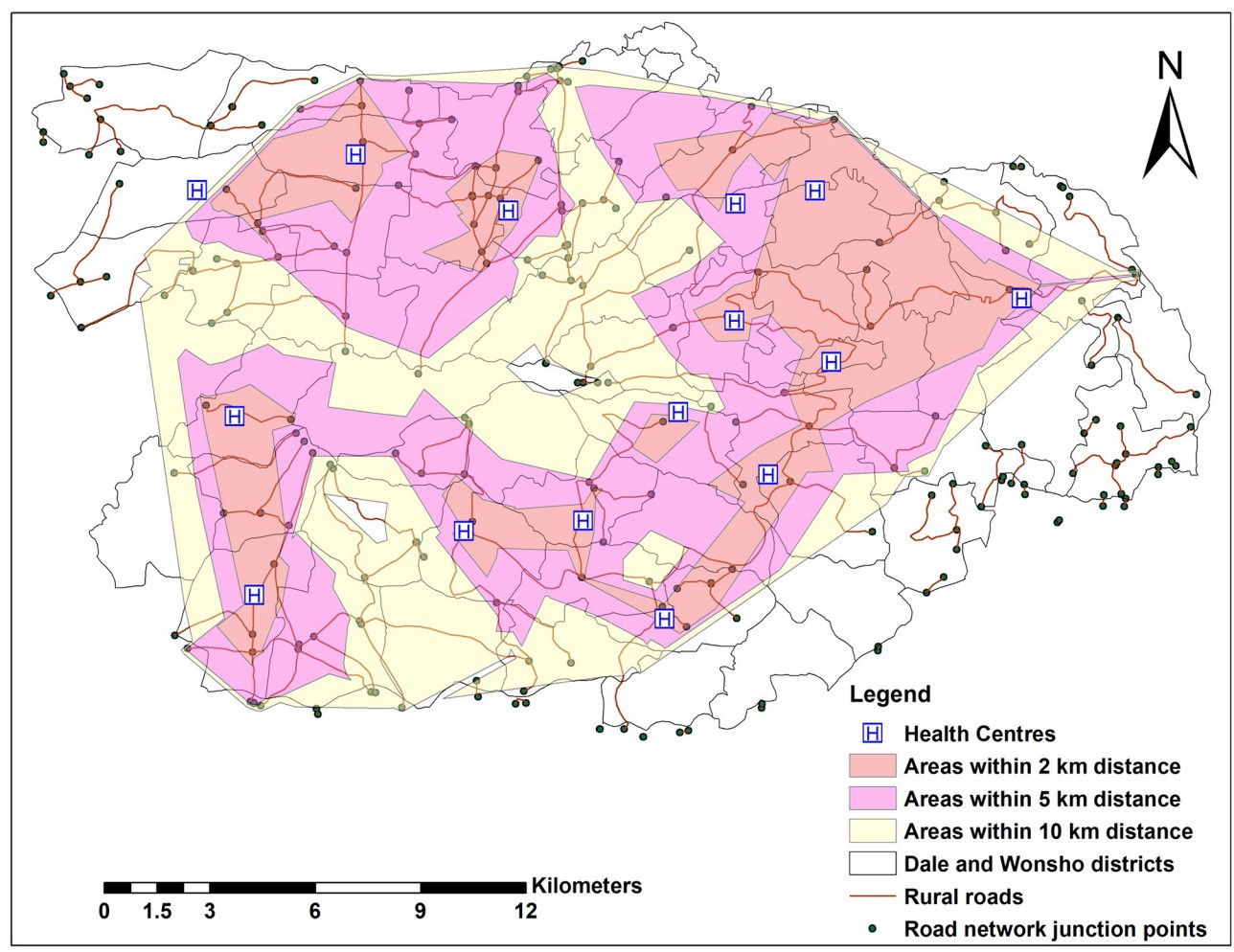

**Fig 3. Distribution of health centres and distance of areas away from the health centres based on the road networks in Dale and Wonsho districts, Sidama, Ethiopia, 2017/18.**

in the most likely cluster and secondary cluster remained the same as unadjusted purely spatial analysis, except one additional kebele from the Wonsho district was incorporated as a secondary cluster (Fig 5 and S3 Table).

The linear regression analysis showed an inverse relationship between distance from the health centre to the kebele central location and the health service utilisation rate (b-estimate = -0.05, p-value = 0.02, 95% CI (-0.08, -0.01)). This means, for every 1 km increase in mean

**Table 1. Most likely and secondary spatial clusters of low health service utilisation detected by purely spatial analysis in Dale and Wonsho districts, Sidama, southern Ethiopia, 2017/18.**

| Cluster | Number of cluster locations | Observed cases | Expected cases | Relative risk | Likelihood ratio | P_value |
|---|---|---|---|---|---|---|
| Most likely cluster | 23 | 23572 | 31663.50 | 0.74 | 1982.93 | 0.001 |
| Secondary cluster | 1 | 255 | 733.06 | 0.35 | 210.49 | 0.001 |
| Secondary cluster | 2 | 1385 | 2227.62 | 0.62 | 189.82 | 0.001 |
| Secondary cluster | 1 | 1381 | 1674.93 | 0.82 | 28.10 | 0.001 |
| Secondary cluster | 1 | 1395 | 1558.16 | 0.90 | 9.06 | 0.004 |

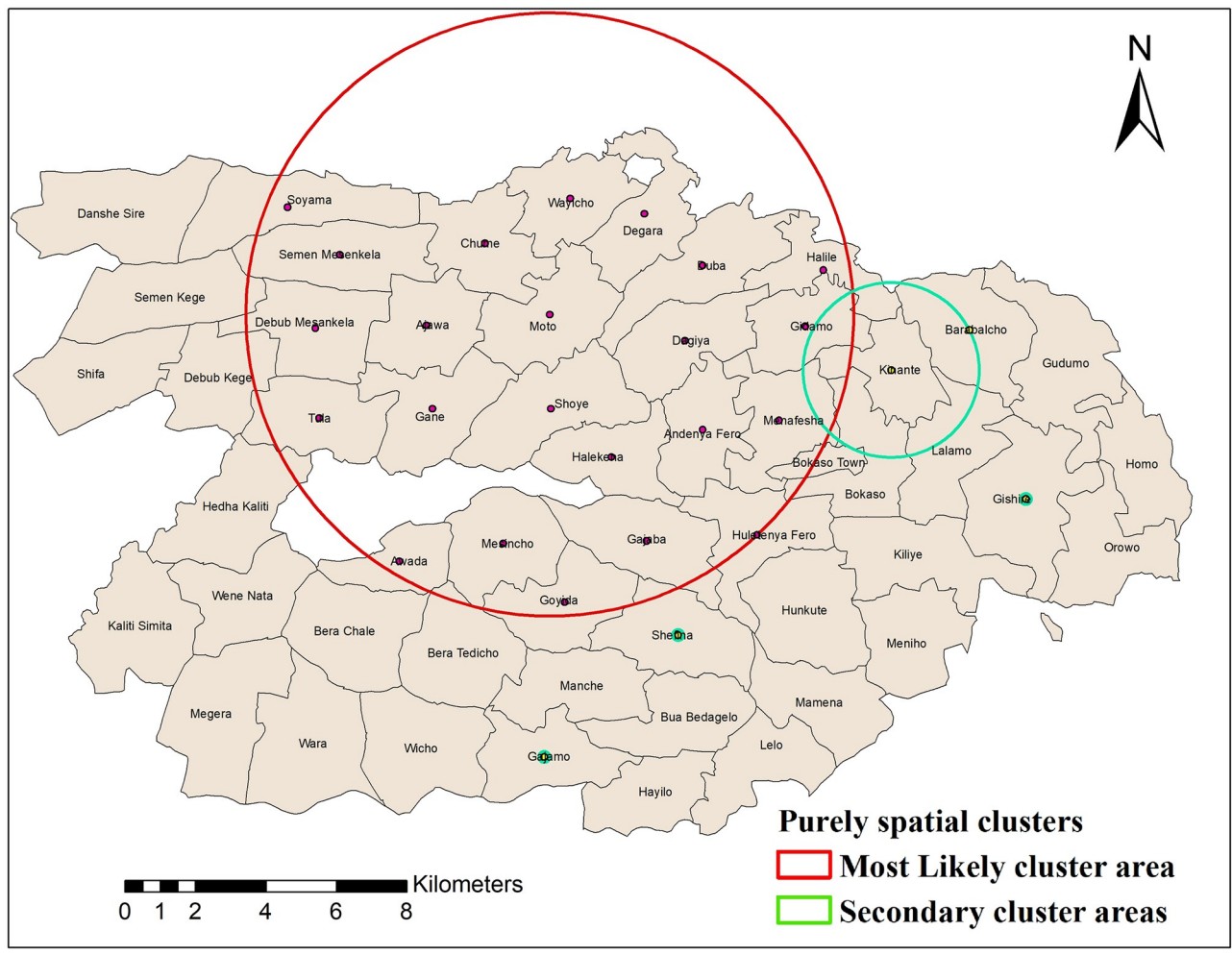

**Fig 4. Purely spatial clustering of low health service utilisation in Dale and Wonsho districts, Sidama, Ethiopia, 2017/18.**

distance from the nearest health centre, the health service utilisation rate decreased by an average of 0.05 visits per 1000 people (S4 Table).

## Discussion

We found that the health service utilisation rate was not randomly distributed at the kebele level in the Dale and Wonsho districts. Urban areas and kebeles with a health centre had higher health service utilisation rates compared with rural kebeles and kebeles that do not have a health centre. This might be due to proximity or better physical accessibility of health centres to urban areas and the availability of services on all days of the week at the health centres, unlike the health posts.

In the previous study done by the same authors, a low health service utilisation rate was identified in the study area. However, urban populations used health services more than rural people. The current study confirmed that higher rates of health service utilisation concentrated around an urban kebele. Areas far away from health centre locations had lower health service utilisation rates.

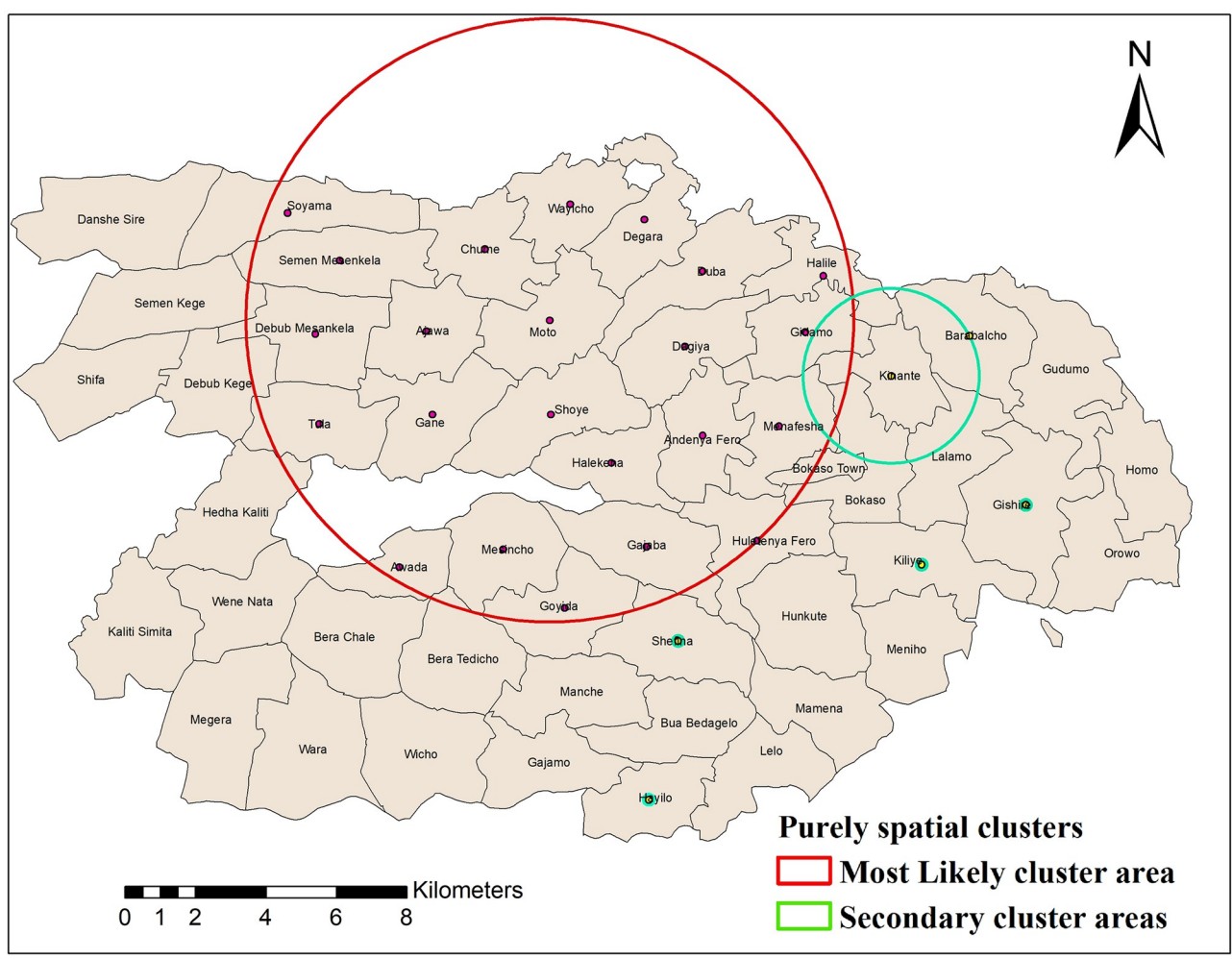

**Fig 5. Covariate adjusted spatial clustering of low health service utilisation in Dale district, Sidama, Ethiopia, 2017/18.**

One of the strengths of this study was including all geographical centres of the kebeles in both districts to assess the health service utilisation rate. Similarly, performing analysis at the lowest scale minimizes the ecological bias. Including available kebele level variables for covariate adjustment and multivariable regression analysis has ruled out some area-level determinants for the low health service utilisation.

Our study has potential limitations that arise from the methodology we used. First, the nature of this exploratory study couldn't identify a causal relationship. Second, we didn't collect data for each individual but at the kebele level. However, kebeles are the lowest spatial units involved in health care planning through their administrative and community representatives in Ethiopia [39]. Third, the health facility registry data lacks comprehensive information to assess determining factors for clustering of health service utilisation. For this reason, our analysis was limited to the identification of clustering and some geographic determinants only. Moreover, the quality of data from the health service information system might be poor [40]. However, since we used all registries from all health centres and health posts from Dale and Wonsho districts, there might not be selective information bias. An ecological bias that arises from the aggregation of data is another concern. However, we performed the analysis at

the lower administrative level which is closer to the individual level which reduces the ecological bias [41]. Besides, most of the exposure variables included in this study, altitude, population density, and distance from the health centre to the centre of the kebele, were unlikely to be different at kebele and individual levels. This implies that the findings observed at kebele level holds true at individual level as well. As we did not collect data from hospitals, private health facilities and for cases that sought services outside the study area, this might result in the underestimation of cases in kebeles closer to these facilities or Yirga Alem town, where the hospital and other private clinics are located. However, the population in those kebeles were around 6% of the study area (23,313/398,569 people) and may not change the utilisation rate.

We also did not consider other individual and biological factors that could contribute to the aggregation of health services in this study. Hence, we refrained from making inferences at an individual level to avoid the ecological fallacy. Furthermore, since there was no census conducted in recent years, we used the 2007 census projection estimate for the current population size in the study area, which might result in an inaccurate denominator estimate for our calculations.

Our study showed better health service utilisation at urban kebeles. This finding is consistent with different studies conducted in Ethiopia [13, 21, 42, 43]. Contrary to our finding, a study done in Kenya found that urban dwellers utilising health care services less than rural residents [44]. This inconsistency might be due to the societal difference portrayed by the Kenyan study as having a habit of self-medication by the over-the-counter drugs, in addition to the differences in the study design.

Kebeles with better access to health centres had a higher health service utilisation rate than other kebeles that didn't have a health centre. This finding is similar to other community-based studies done in Ethiopia that depicted people living close to health centres or hospitals as having a better health service utilisation [22, 43]. Similarly, a recent study from the same study area showed that people preferred to use health centres over health posts [34]. Therefore, the government's initiative to scale up the health posts to a level in which they can give good health service with better-trained professionals and equipments, is in line with our findings [17]. Hence, we believe that changing the health system from basic to comprehensive health post level might improve health service utilisation.

Constructing health centres within a 5 km distance was reported to increase service delivery significantly in Ethiopia [45]. Similarly, another study in Amhara region showed that access to health services was affected by the lower health service coverage and geographically inaccessible location of health facilities [46]. The national standard also states a physically accessible health facility if it is within a 10 km or 2 hours walking distance. In our study, we found that more than half (57.4%) of the locations were within a 10 km distance from the health centres. One of the explanations for this finding might be the exclusion of the 50 health post locations from our analysis, which might have increased the coverage substantially. Yet, the recommendation for the construction of health centres within a 5 km reach was not achieved in our study area. Our study might give insight into geographic areas in which people lack access to better health services. Furthermore, this finding implies constructing health institutions nearer to the homes to improve accessibility as a proxy for better service usage. Besides, this result supports the expansion and upgrading of health facilities in Ethiopia to meet the population needs, as indicated by the new Ministry of Health initiative to optimise the health primary health care units [17].

Different studies in Ethiopia found that geographical variables, such as distance from the health facilities, altitude and population density as determinant factors for clustering of specific diseases [13, 22, 35]. In this study, we only found the distance from the health centre having an inverse relationship with the health service utilisation rate. The geographic similarity of the

adjacent districts we studied might have contributed to the non-significant finding on altitude and population density. Moreover, in our study, we used institution-based study taking measurements from the kebeles' central location while the other studies used individual or household level measurements.

## Conclusions

This study showed considerable geographical variability in terms of health service utilisation in the study area. The geographical accessibility of the health centres significantly affected the health service utilization rate. Therefore, our findings support the government's initiative of upgrading the primary health care units to improve the physical accessibility and the health services to the rural community. Moreover, understanding the variations and investigating other geographical factors responsible for this aggregation could give some valuable insights to the health planners in allocating limited resources and devising targeted interventions specific to the local areas.

## Supporting information

**S1 Table. List of registers reviewed to extract the data from health centres and health posts in Dale and Wonsho districts, 2017/18, Sidama, Ethiopia.** NB: The information extracted from each registers were, name of the district, name of health facility, address of the health facility (rural or urban), unit or department, card number, date of visit, sex, age (in days, months, years or date of birth), address (kebele), diagnosis, and visit type.
(DOCX)

**S2 Table. The annual health service utilisation rate of new cases in kebeles of Dale and Wonsho districts, 2017/18, Sidama, Ethiopia.**
(DOCX)

**S3 Table. Most likely and secondary spatial clusters of low health service utilisation detected after adjustment of covariates in purely spatial analysis in Dale and Wonsho districts, Sidama, southern Ethiopia, 2017/18.**
(DOCX)

**S4 Table. Multivariable linear regression analysis of geographic factors affecting the health service utilisation in Dale and Wonsho districts, Sidama, southern Ethiopia, 2017/18.** We did the analysis using aggregated data from 54 kebles (n = 54 kebeles), R-squared = 0.15, P-value = 0.04.
(DOCX)

**S1 Fig. Conceptual framework based on the behavioral model of health service use by Ronald M. Andersen, 1995.**
(TIF)

**S2 Fig. The areas within 10 km distance coverage of the health centres in Dale and Wonsho district, Sidama, Ethiopia, 2017/18.**
(TIF)

## Acknowledgments

We would like to acknowledge University of Bergen and Hawassa University for supporting this study. We are also grateful for Sidama Regional Health Bureau, Dale and Wonsho district health offices, Sidama Region Plan Commission and Central Statistical Agency, Hawassa

branch, and personnel at the health facilities in Dale and Wonsho districts for their cooperation. We wish to thank all our data collectors and supervisors for their involvement in this study.

## Author Contributions

**Conceptualization:** Hiwot Abera Areru, Mesay Hailu Dangisso, Bernt Lindtjørn.

**Data curation:** Hiwot Abera Areru, Mesay Hailu Dangisso, Bernt Lindtjørn.

**Formal analysis:** Hiwot Abera Areru, Mesay Hailu Dangisso, Bernt Lindtjørn.

**Funding acquisition:** Bernt Lindtjørn.

**Investigation:** Hiwot Abera Areru, Mesay Hailu Dangisso, Bernt Lindtjørn.

**Methodology:** Hiwot Abera Areru, Mesay Hailu Dangisso, Bernt Lindtjørn.

**Project administration:** Hiwot Abera Areru, Mesay Hailu Dangisso, Bernt Lindtjørn.

**Resources:** Hiwot Abera Areru, Mesay Hailu Dangisso, Bernt Lindtjørn.

**Software:** Hiwot Abera Areru, Mesay Hailu Dangisso, Bernt Lindtjørn.

**Supervision:** Hiwot Abera Areru, Mesay Hailu Dangisso, Bernt Lindtjørn.

**Validation:** Hiwot Abera Areru, Mesay Hailu Dangisso, Bernt Lindtjørn.

**Visualization:** Hiwot Abera Areru, Mesay Hailu Dangisso, Bernt Lindtjørn.

**Writing – original draft:** Hiwot Abera Areru, Mesay Hailu Dangisso, Bernt Lindtjørn.

**Writing – review & editing:** Hiwot Abera Areru, Mesay Hailu Dangisso, Bernt Lindtjørn.

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
