## [Decision Letter · Decision Letter 0]

4 Oct 2021

PGPH-D-21-00546

Large local variations in the use of health services in rural southern Ethiopia: an ecological study

Dear Dr. Areru,

Thank you for submitting your manuscript to PLOS Global Public Health. After careful consideration, we feel that it has merit but does not fully meet PLOS Global Public Health’s publication criteria as it currently stands. Therefore, we invite you to submit a revised version of the manuscript that addresses the points raised during the review process.

We look forward to receiving your revised manuscript.

Kind regards,

Mark S. Pearce, PhD

Academic Editor

Journal Requirements:

1. Please provide us with a direct link to the base layer of the map used in 2-5 and ensure this location is also included in the figure legend. 

Please note that, because all PLOS articles are published under a CC BY license (creativecommons.org/licenses/by/4.0/), we cannot publish proprietary maps such as Google Maps, Mapquest or other copyrighted maps. If your map was obtained from a copyrighted source please amend the figure so that the base map used is from an openly available source.

Please note that only the following CC BY licences are compatible with PLOS licence: CC BY 4.0, CC BY 2.0  and CC BY 3.0, meanwhile such licences as CC BY-ND 3.0 and others are not compatible due to additional restrictions. If you are unsure whether you can use a map or not, please do reach out and we will be able to help you. 

The following websites are good examples of where you can source open access or public domain maps:

Reviewers' comments:

Reviewer's Responses to Questions

**Comments to the Author**

1. Does this manuscript meet PLOS Global Public Health’s publication criteria? Is the manuscript technically sound, and do the data support the conclusions? The manuscript must describe methodologically and ethically rigorous research with conclusions that are appropriately drawn based on the data presented.

Reviewer #1: Yes

Reviewer #2: Partly

Reviewer #3: Yes

2. Has the statistical analysis been performed appropriately and rigorously?

Reviewer #1: Yes

Reviewer #2: I don't know

Reviewer #3: Yes

3. Have the authors made all data underlying the findings in their manuscript fully available (please refer to the Data Availability Statement at the start of the manuscript PDF file)?

Reviewer #1: Yes

Reviewer #2: Yes

Reviewer #3: Yes

4. Is the manuscript presented in an intelligible fashion and written in standard English?

Reviewer #1: Yes

Reviewer #2: Yes

Reviewer #3: Yes

5. Review Comments to the Author

Reviewer #1: Review

The authors conducted an ecological study to demonstrate that utilisation rates varied by geographical access in rural southern Ethiopia

They have employed sound methods.

The style of writing is acceptable

A few comments will help make the paper better

1. Abstract. The author should state explicitly that Secondary data was used. They should state the number of Kebeles studied. The variation of the coverage or visits/ population / perpetuation and an overall mean or median.

2. Introduction. There is the need to rearrange some part of this. The area dealing with use of geographic analysis should come only after the main problems of coverage has been put perspective. Reference 10 is rather 30years old. The research question should be more pointed and the justification. Much of the part informing readers of Ethiopia’s plans to improve its health system may not be useful and not relevant

3. Methods were sound but it should be stated that Secondary data were used and that the study sites were purposive chosen.

4. Results. Line 266, the RR values for the most likely cluster should be cross checked.

5. Discussion. Line 306. The authors should show why gave think that there are no variations between household legal and kebele level. The implications of the fi dings have not been discussed adequately

Reviewer #2: It is well known that health care availability and geographic access are necessary preconditions to universal coverage in health. From that point of view, the paper did not provide new evidence to the field. However, the methodological approach, the ecological analysis, the incorporation of geographical information systems, and the creativity to overcome the scarce individual data are valuable and could help other research in similar contexts.

Despite this, the manuscript will require revision to clarify many issues, beginning with the health access conceptual framework and definitions for coverage and access. The authors must provide background information to understand the context, including demographic composition, epidemiological profile, and more detail about the health care facilities resources and registry system. The discussion section requires to be connected with the health care access conceptual framework.

Specific comments:

Line 55: Please define coverage and access. It requires a definition of coverage, access, and utilization.

Line 57: Clarify that this is a national definition said afterward, but this is the first mention.

Line 65: Explain the differences between health posts and health centers.

Line 103 Please add a little detail about additional health determinants such as % women, % of children, poverty. And if it is possible leading causes of death and morbidity.

Line 132: Provide more detailed pieces of information about the selected centers' registries and their integrity or quality. Specify the way the authors acceded to those registries. Provide the rationale to include only new cases; this should be related to the definition of health care utilization in the conceptual framework.

Line 141 Explain the utilization of the 1.5 design effect.

Line 152: Give more detail of the data collection instrument to the health care registries system and how the authors acceded them.

Line 161: Clarify the operational definition used to measure the variable distance to health centers. It seems that is a proxy.

Line 166: It seems that the authors used the distance to health centers and not to health posts, clarify.

Line 189: Define service area coverage.

Reviewer #3: This is a very well written and clear manuscript of an ecological study assessing the variation in health service utilisation in rural southern Ethiopia. I spotted a couple of minor typos (line 191 missing the word 'was' used to ... and line 317 4th word 'that' should be removed). I would like further detail on what constitutes 'family size', this is an unfamiliar term to me (perhaps an example added to the text would suffice). Otherwise the analysis method is well considered and explained, and the conclusions well supported and justified. An enjoyable read, I hope this work has impact to support improvement in health service accessibility.

6. PLOS authors have the option to publish the peer review history of their article (what does this mean?). If published, this will include your full peer review and any attached files.

**Do you want your identity to be public for this peer review?** For information about this choice, including consent withdrawal, please see our Privacy Policy.

Reviewer #1: No

Reviewer #2: No

Reviewer #3: **Yes: **Dr Kay Mann

---

## [Decision Letter · Decision Letter 1]

1 May 2022

Large local variations in the use of health services in rural southern Ethiopia: an ecological study

PGPH-D-21-00546R1

Dear Mrs Hiwot Abera Areru

We are pleased to inform you that your manuscript 'Large local variations in the use of health services in rural southern Ethiopia: an ecological study' has been provisionally accepted for publication in PLOS Global Public Health.

Best regards,

Alinane Linda Nyondo-Mipando, PhD

Academic Editor

Reviewer Comments (if any, and for reference):

Reviewer's Responses to Questions

**Comments to the Author**

1. If the authors have adequately addressed your comments raised in a previous round of review and you feel that this manuscript is now acceptable for publication, you may indicate that here to bypass the “Comments to the Author” section, enter your conflict of interest statement in the “Confidential to Editor” section, and submit your "Accept" recommendation.

Reviewer #2: All comments have been addressed

2. Does this manuscript meet PLOS Global Public Health’s publication criteria? Is the manuscript technically sound, and do the data support the conclusions? The manuscript must describe methodologically and ethically rigorous research with conclusions that are appropriately drawn based on the data presented.

Reviewer #2: Yes

3. Has the statistical analysis been performed appropriately and rigorously?

Reviewer #2: I don't know

4. Have the authors made all data underlying the findings in their manuscript fully available (please refer to the Data Availability Statement at the start of the manuscript PDF file)?

Reviewer #2: Yes

5. Is the manuscript presented in an intelligible fashion and written in standard English?

Reviewer #2: Yes

6. Review Comments to the Author

Reviewer #2: (No Response)

7. PLOS authors have the option to publish the peer review history of their article (what does this mean?). If published, this will include your full peer review and any attached files.

**Do you want your identity to be public for this peer review?** For information about this choice, including consent withdrawal, please see our Privacy Policy.

Reviewer #2: No
